# The Effects of Stress and Diet on the “Brain–Gut” and “Gut–Brain” Pathways in Animal Models of Stress and Depression

**DOI:** 10.3390/ijms23042013

**Published:** 2022-02-11

**Authors:** Mauritz F. Herselman, Sheree Bailey, Larisa Bobrovskaya

**Affiliations:** Health and Biomedical Innovation, Clinical and Health Sciences, University of South Australia, Adelaide, SA 5000, Australia; mauritz.herselman@mymail.unisa.edu.au (M.F.H.); sheree.bailey@unisa.edu.au (S.B.)

**Keywords:** animal, depression, diet, gut–brain axis, microbiota, pathways, prebiotics, probiotics, stress

## Abstract

Compelling evidence is building for the involvement of the complex, bidirectional communication axis between the gastrointestinal tract and the brain in neuropsychiatric disorders such as depression. With depression projected to be the number one health concern by 2030 and its pathophysiology yet to be fully elucidated, a comprehensive understanding of the interactions between environmental factors, such as stress and diet, with the neurobiology of depression is needed. In this review, the latest research on the effects of stress on the bidirectional connections between the brain and the gut across the most widely used animal models of stress and depression is summarised, followed by comparisons of the diversity and composition of the gut microbiota across animal models of stress and depression with possible implications for the gut–brain axis and the impact of dietary changes on these. The composition of the gut microbiota was consistently altered across the animal models investigated, although differences between each of the studies and models existed. Chronic stressors appeared to have negative effects on both brain and gut health, while supplementation with prebiotics and/or probiotics show promise in alleviating depression pathophysiology.

## 1. Introduction

Depression is becoming increasingly prevalent as the world modernizes and is projected to be the number one health concern by 2030, which may be in part due to its close association with stress [1,2,3,4]. At a global scale, it affects approximately 350 million people, and it is the leading cause of disability bringing about substantial economic burden [1,5,6]. Such disability being a result of the characteristics of the disorder including cognitive, emotional, memory and motor function impairment [2].

Several different hypotheses of depression have been developed, covered extensively in other reviews [2,7]. It is also well established that overuse or dysregulation of the stress response in mammals promotes depression pathophysiology [7]. Recently, compelling evidence is building for the involvement of the complex, bidirectional communication axis between the gastrointestinal tract and the central nervous system (CNS) which serves to mediate mood states, in turn affecting intestinal motility, secretion and immune function [8,9]. The gut microbiota, the conglomerate of symbiotic or pathologic microorganisms which reside in the gastrointestinal tract, such as bacteria, fungi, viruses and archaea, form a key part in this line of communication [5,8].

From the brain to the gut, stress affects multiple pathways, beginning with the perception of stress by the frontal cortex of the brain, leading to activation of the major stress response pathways: the hypothalamic–pituitary–adrenal (HPA) axis and sympatho–adrenomedullary system [2,7]. In addition to producing glucocorticoids, the HPA axis regulates multiple central and peripheral pathways including neurogenesis, immune function, and tryptophan metabolism which in turn affects serotonin availability to the CNS [2,7]. Activation of the sympatho–adrenomedullary system by stress results in the release of catecholamines, such as adrenaline and noradrenaline, by the adrenal glands into the circulation which may exert effects on gut function [10]. Acting as a brake on these sympathetic driven responses on gut function, the vagus nerve, which largely contributes to the parasympathetic nervous system, acts efferently to decrease intestinal inflammation and strengthen intestinal barrier function [11,12]. These efferent fibres of the vagus nerve are inhibited by stress, allowing the sympathetic effects on the gut to continue unchecked [11,12]. From the gut to the brain, compromised intestinal barrier integrity brought about by stress allows enteric bacteria and their associated ligands to enter the circulation, leading to systemic inflammation with further consequences for brain health [13]. Afferent fibres of the vagus nerve present in the gut send signals to the brain, relaying the status of the intestinal environment and regulate the HPA axis accordingly [12]. Thus, the bidirectional communication that exists between the gut and the brain suggests that diets may have powerful effects on the modulation of the microbiota and the gut–brain axis, with implications for brain health. A review by Taylor and Holscher [14] found dietary interventions in depression were able to improve mood and reduce depression scores. Observational and animal studies have shown diets high in fibre and nutrients, but low in refined carbohydrates and saturated fat, such as the Mediterranean or Japanese diets, have positive effects on mental health and may prevent the onset of depression [15,16]. Therefore, due to the complex multi-organ systems involved in the gut–brain axis, animal models of stress are a viable tool to investigate the bidirectional communication along this axis as well as the effects of diets on its pathways. This review will evaluate recent studies which used common animal models of stress and depression to investigate the effects of stress and diets on the brain-to-gut and gut-to-brain pathways.

## 2. Animal Models of Stress and Depression: An Overview

Modelling a complex human disorder such as depression in animals is a challenging endeavour due to the heterogeneity in symptoms and complexity of the biomolecular systems involved. Thus, current animal models of depression described next rely on criteria where the model must produce a similar behavioural phenotype, respond to clinically effective antidepressant treatments and should have a comparable neurobiological phenotype; however, there are limitations to the currently established animal models that may not reflect the heterogeneity in the biological systems and behavioural symptoms between depressive patients [1,17]. Considering this, studies using animal models of stress require a range of measures for stress-induced behaviours that manifest in clinical depression. These include, measuring the preference for a sweet substance in rodents to characterize anhedonia (akin to the inability to feel pleasure in humans), using spatial memory tasks to measure cognitive impairment, assessing grooming behaviour to gauge feelings of worthlessness and measuring durations of social interactions to determine dysfunctional social behaviour [1,18,19]. Anxiety-related behaviour and alterations in locomotor activity are also frequently measured using open field or elevated plus maze tests, and behavioural despair or passive coping are also commonly measured using forced swimming or tail suspension tests, although it is unclear how these two latter paradigms translate to human depression symptomology. Recently, the ethics and validity of the tail suspension test and forced swimming test which measure behavioural despair have been questioned, creating the need for novel behavioural tests to better reproduce the human condition [20].

Most animal models of stress and depression also use male rodents, whilst female studies are sparser. This gender bias is usually employed to avoid potential effects of the oestrous cycle on parameters such as behaviour and molecular mechanisms, despite depression rates being higher in women [21]. Additionally, female studies often use stress models that were optimised in male rodents which may not effectively capture the female stress response. Female rodents are more social and sensitive to isolation whereas males are territorial and exhibit same-sex aggression [22]. Since sex differences exist in the current stress models, female-specific behaviour and stress effects will be highlighted in this review when the studies used female rodents.

Decades of pre-clinical research related to brain and behaviour has resulted in the establishment of several different commonly used animal models of stress and depression which include maternal separation, restraint stress, exogenous glucocorticoid administration, social defeat stress, learned helplessness and chronic unpredictable mild stress [6]. Other animal models of depression have also been established such as olfactory bulbectomy, however such models are controversial in their effects and require specialized techniques [6]. Thus, this review will focus on models which are most commonly used to model chronic stress and depression.

Maternal separation (MS), typically referred to as early life stress, is based on the premise that the development of depression is often related to adversity in early life [6,23]. Thus, the MS model utilizes postnatal separation of pups from their mothers for several hours or several days and this has been shown to successfully dysregulate the HPA axis [6]. Dysregulation of the HPA axis is well established to be a core feature of depression pathophysiology, leading to chronically elevated glucocorticoid levels through impaired negative feedback via glucocorticoid receptors in the hypothalamus and pituitary, which may also be a cause of hippocampal and prefrontal cortex apoptosis [6]. However, it is also known that hypoactivation of the HPA axis can occur in depression, particularly in females or in individuals with atypical depression [24]. This is likely due to the complex differential effects of sex hormones on HPA axis function which has been demonstrated in MS models of depression where male rats appear to be more susceptible to exhibiting HPA hyperreactivity to stress compared with female rats [25,26].

Chronic restraint stress (CRS) induces depressive-like behaviours by means of restraining animals daily for a few hours over a period of 14 to 21 days [6]. While outcomes are known to vary with this model, it is generally accepted that this model results in atrophic CA3 pyramidal cells, which play a specific role in memory, in the hippocampus and elevated levels of corticosterone in stressed animals. A study comparing the effect of CRS in male and female rats demonstrated that atrophic CA3 pyramidal cells were present in males and absent in females. Differing corticosterone responses were discounted as a cause for the difference and thus require further study [27].

Chronic unpredictable mild stress (CUMS) involves the application of several mild stressors, including restraint stress, food and water deprivation, cage tilt and others, over a number of weeks in order to bring about depressive-like behaviours such as anhedonia, which this model is specifically well suited for [6,28,29]. CUMS is widely known to be robust in dysregulating HPA axis function as well as adversely affecting brain regions involved in depression pathophysiology such as the hippocampus and prefrontal cortex [6]. This model is likely more suitable to model depression in female mice since its chronic nature exposes females to stressors during all phases of the oestrous cycle [30]. However, behavioural outcomes from this model show differential effects of sex, since male rodents show a different consumption pattern compared to females when measuring intake of a sweet substance, with females showing a more gradual reduction in intake; nonetheless anhedonia is still ultimately produced in both sexes, although more pronounced in females [22,30]. Females also tend to exhibit higher levels of exploratory behaviour compared to males in the open field and elevated plus maze tests, and spend less time immobile in the forced swimming test, suggestive of lower anxiety and behavioural despair, but female rodents are also known to exhibit enhanced reactivity to potential threats, thus it may be that traditional behavioural tests do not sufficiently capture forms of female anxiety-like behaviour [22,30,31].

Other models of stress and depression include chronic social defeat stress (CSDS) whereby young male animals are placed into an older dominant male’s home cage to engage social conflict in order to bring about anhedonia and anxiety-like behaviours; learned helplessness (LH) whereby animals are briefly subjected to an inescapable shock at random to produce anxiety-like and helpless behaviours; and the exogenous administration of glucocorticoid, all of which have been demonstrated to dysregulate HPA axis function [6,32,33,34]. Sex differences in these latter models have been more difficult to establish since CSDS relies on male aggression and differences found in LH are not reversed by gonadectomy [35]. 

Overall, female rodents appear less susceptible to depression across current animal models of depression in contrast to humans. According to Scholl et al. [31], this may be due to differing coping strategies in each sex or it may be that stress-related behaviours manifest differently in females compared to males, accentuating the need for female-specific behavioural tests [22]. Yet, biologically, alterations in serotonergic activity and HPA axis function in females suggest females are in fact more vulnerable to the effects of stress in rodents [31].

Recent research has shown that the composition of the gut microbiota is altered in the animal models of stress and depression described, however consistency amongst reported changes and the likelihood of stress and depression as the cause is not well established. Thus, we will review the effects of stress on the bidirectional connections between the brain and the gut across the most widely used animal models of stress and depression, followed by comparisons of stress-induced alterations to the gut microbiota with possible implications for the gut–brain axis and the impact of dietary changes on these.

## 3. The Effects of Stress on the Bidirectional Pathways between the Brain and the Gut and Their Possible Contribution to Depressive Behaviours

From the brain to the gut, exposure to stress activates the major stress response systems (HPA axis and neural pathways such as the sympatho–adrenomedullary system of the autonomic nervous system) which together modulate the intestinal environment by increasing colonic inflammation through inhibition of the vagus nerve, decreasing intestinal mucus and epithelium integrity [23,28,29], and altering the composition of the gut microbiota (see Section 3.2.2. Gut Microbiota Pathways). Dysbiosis of the gut microbiota further decreases intestinal barrier integrity due to a reduction in the availability of microbiota-produced-short chain fatty acids (SCFA) [23,28,29]. From the gut to the brain, this leads to exposure of the circulation to bacteria and their products which activate inflammatory mechanisms [29,32,33,34,36,37] and increase output of the neurotoxic branch of the kynurenine pathway [28,33,38,39,40] which act synergistically to increase neuroinflammation inducing depressive- and anxiety-like behaviours. These pathways are summarised in Figure 1 and all discussed in more detail in the sections to follow.

### 3.1. Brain-to-Gut Pathways

#### 3.1.1. Neuroendocrine Pathways: HPA Axis

The HPA axis, a major neuroendocrine system involved in the stress response, regulates glucocorticoid production and influences digestion, the immune system, mood and behaviour. Dysregulation of the HPA axis and increased inflammation have been highlighted as mechanisms for stress-induced behaviour changes [36,41]. It was demonstrated that chronic stress overproduced serum corticosterone in MS [23] and CUMS male mice, and elevated levels of hypothalamic corticotropin-releasing factor (CRF) in association with increased depressive- and anxiety-like behaviours [42]. The impaired negative feedback of corticosterone in the HPA axis was attributed to downregulation of the glucocorticoid receptor, existing with increased levels of inflammatory cytokines; serum TNF-α and hippocampal IL-1 [42]. In female mice, CUMS similarly overproduced serum corticosterone in association with increased depressive- and anxiety-like behaviours [43].

Even so, according to Bonaz et al. [11], CRF receptors 1 and 2 are expressed in both the brain and the gastrointestinal tract and may modulate the gut microbiota as well as increase intestinal barrier permeability via TNF-α release from mast cells in the gut which also express CRF receptors, suggesting that the HPA axis may influence the gut–brain axis (Figure 2). Moreover, there is evidence that CRF synthesis also occurs in enteric neurons and enterochromaffin cells under inflammatory conditions [10]. Liu et al. [33] showed that CRS increases colonic CRF peptide in conjunction with colonic motility and mucus secretion in male rats, although this was not accompanied by behavioural data. These CRS-induced effects as well as increased numbers of CRF-immunoreactive neuronal cell bodies in the myenteric and submucosal plexuses were not found in non-stressed control rats, and knockdown of colonic CRF expression was further shown to prevent the CRS-induced effects on colonic motility and mucus secretion [33]. Amini-Khoei et al. [23] showed that dysregulated HPA axis function in MS existed with increased colonic inflammation, necrosis and shedding of the intestinal epithelium, however their study did not determine whether such changes were directly modulated by CRF receptor signalling. Interestingly, Vodička et al. [32] showed CSDS had no effect on colonic CRF expression, but that colonic levels of the ubiquitously expressed enzyme responsible for the conversion of inactive deoxycorticosterone to active corticosterone, 11β-hydroxysteroid dehydrogenase 1 (11βHSD1), was decreased while expression levels were higher in germ free mice in the same paradigm, indicating modulation of local glucocorticoid regulation by the gut microbiota. Colonic glucocorticoid synthesis, which does not respond to adrenocorticotropic hormone, is considered separate to adrenal glucocorticoid synthesis (see the reviews by Ahmed et al. [34], Kostadinova et al. [37]) and may contribute to the maintenance of intestinal barrier integrity as well as intestinal immune homeostasis. In fact, evidence indicates local glucocorticoids have an antagonistic effect on TNF-α-dependent decreases in intestinal barrier integrity, suggesting that the decrease in local glucocorticoid synthesis brought about by CSDS reported by Vodička et al. [32] would have contributed to compromised intestinal barrier integrity [34]. Overall, these findings indicate that the stress response, via the HPA axis, may have significant consequences for gut health in animal models of stress and depression, therefore potentially impacting other pathways since stress affects multiple central and peripheral pathways including neuroplasticity, tryptophan metabolism and immune function [2,7]. Furthermore, the neuroinflammation in the reported studies associated with chronic activation of the HPA axis may be due to impaired intestinal barrier integrity inducing systemic inflammation and exposing the brain to inflammatory cytokines (see Section 3.2.1 Intestinal Barrier Integrity Pathways). Although beyond the scope of this review, this vulnerable state of the brain during stress is brought about by direct modulation of the blood–brain barrier by inflammatory cytokines derived from the central immune system and microglia, a process well-described in the review by Doney et al. [44]. Another explanation may be due to increased neurotoxicity driven by increased output from the neurotoxic branch of the kynurenine pathway (see Section 3.1.3 Serotonin and Tryptophan Pathways). Nonetheless, neural pathways involved in the stress response are pivotal in the overall bidirectional communication between the brain and the gut.

#### 3.1.2. Neural Pathways

Exposure to stress activates the autonomic nervous system (ANS) which exerts control over the sympatho–adrenomedullary system and enteric nervous system (ENS) [10]. Activation of the sympatho–adrenomedullary system results in the release of catecholamines, such as adrenaline and noradrenaline, by the adrenal glands into the circulation, which act on several different organs, but also reach the gut via sympathetic nerve terminals, where they may influence the composition of the gut microbiota (Figure 3) [10]. In support of this, in vitro studies have also shown Gram-negative bacteria have a log-fold increase in growth when exposed to noradrenaline [10]. Moreover, in recent studies, adrenal enzymes involved in catecholamine biosynthesis, such as tyrosine hydroxylase and phenylethanolamine *N*-methyltransferase (PNMT), were increased under CUMS or CSDS and have shown links to the gut microbiota since germ free mice exhibited higher adrenal levels of these key enzymes, although the mechanisms involved are currently unclear [32,36].

The ANS also exerts control over the enteric nervous system (ENS), located in the myenteric plexus and mucosa of the gut, which contains adrenergic receptors that upon activation may influence the luminal environment of the gut, further modulating the gut microbiota by altering intestinal permeability and motility [10]. In fact, the ENS consists of approximately 20 different neuronal types, with intestinal permeability, motility and intestinal immune responses predominantly regulated by cholinergic, nitrergic and intrinsic neurotransmission [38,39]. Chronic stress has been shown to increase cholinergic neurons and decrease nitrergic neurons in the ENS of male rats, contributing to increased intestinal permeability and motility [38,40]. Intrinsic nerves of the ENS are also thought to modulate colonic immune responses via substance P, by potentiating inflammation [39,45]. Thus, the activation of such intrinsic nerves of the ENS is thought to be ‘pro-inflammatory’ [39]. Intrinsic nerves of the ENS also contain dopaminergic neurons, which inhibit intestinal motility, thus these effects may contribute to perturbations in gut function brought about by chronic stress [46].

However, control over the ENS is largely mediated parasympathetically via the vagus nerve, which consists of 20% efferent brain-to-gut fibres and 80% afferent gut-to-brain fibres, serving as the main interface between gut and brain homeostatic control [12,13]. These vagal connections act to suppress gut inflammation, but also respond to increases in circulating cytokines, via the dorsal medulla in the brain, and alter vagal motor activity [12]. Chronic stress is known to inhibit the anti-inflammatory potential of the vagus nerve as well as increase circulating proinflammatory cytokines which can reach the brain, therefore increasing gut inflammation and motility [11]. Moreover, Breit et al. [12] described in their review that mediation of the vagal anti-inflammatory potential occurs through the HPA axis, inhibition of TNF-α release via the splenic sympathetic nerve, and the cholinergic anti-inflammatory pathway which inhibits TNF-α release from macrophages via enteric neurons. It is well established that chronic stress dysregulates HPA axis function and increases splenic TNF-α release [47,48]. Furthermore, stress is known to inhibit the vagus nerve, thus increasing TNF-α release from macrophages due to subsequent inhibition of the vagus nerve-mediated cholinergic anti-inflammatory pathway [11]. Stress can thus be considered to have pro-inflammatory effects on the gut [11]. 

Marcondes Ávila et al. [41] demonstrated how crucial vagal connections from the gut are for the brain and behaviour since a faecal microbiota transplant from healthy male donor rats to male rats subjected to CUMS reversed anxiety-like behaviours and decreased oxidative damage and inflammatory IL-6 and TNF-α in the prefrontal cortex and hippocampus; however, these effects, except for the increased cytokines, did not occur in rats that underwent a vagotomy. Thus, while the vagus nerve appears to communicate changes in the gut microbiota to the brain, in turn modulating the behavioural phenotype, chronic stress perceived by the brain inhibits brain-to-gut vagal nerve signalling, promoting intestinal inflammation and motility. These perturbations in gut health allow for the translocation of gut bacteria and their associated products across the intestinal barrier since tight junction proteins of the mucosa are weakened by inflammation [49]. This bacterial translocation drives a systemic inflammatory response, leading to neurotoxicity and depressive-like behaviour through multiple pathways discussed in the sections to follow.

#### 3.1.3. Serotonin and Tryptophan Pathways

Neuroplasticity is well known to be disrupted in depression and is largely modulated by serotonin, through synapse formation [2,6]. Abnormal serotonin signalling in brain regions such as the hippocampus and prefrontal cortex are known to be major pathophysiological mechanisms of depression. Depressive patients are frequently reported to have decreased serotonin metabolites and common antidepressants such as serotonin reuptake inhibitors have been demonstrated to increase serotonin and brain-derived neurotrophic factor (BDNF) in the brain [2,50]. Indeed, several recent animal studies utilized the CUMS paradigm to induce depressive-like and anxiety-like behaviours and showed that these behaviours were associated with decreased serotonin levels in the hippocampus and frontal cortex [42,51,52,53,54].

Serotonin is metabolized from tryptophan, an essential amino acid derived from ingested protein, which is utilized largely by enterochromaffin cells in the gut, but it is also absorbed into the circulation where it can cross the blood–brain barrier [9,55]. Therefore, since serotonin cannot cross the blood–brain barrier, its production in the brain relies on dietary tryptophan reaching the brain where tryptophan hydroxylase (TPH) converts tryptophan into 5-hydroxytryptophan (5-HTP) after which aromatic amino acid decarboxylase produces serotonin from 5-HTP [9]. Yet, 95% of tryptophan is converted separately along the kynurenine pathway in peripheral organs, but especially the gut, which exerts a major influence on serotonin synthesis through the modulation of tryptophan availability, in both the CNS and ENS, the rate of which is under control of indolamine-2,3-dioxygenase (IDO1), which is influenced by inflammatory cytokines (Figure 4) [28,55]. Chronic stress is known to activate the immune system and pro-inflammatory pathways through dysregulation of the negative feedback of the HPA axis by decreasing glucocorticoid receptor expression [2,23]. Under basal conditions, the glucocorticoid-glucocorticoid receptor complex on target immune cells such as macrophages inhibits the transcription factor NF-κB, preventing the expression of proinflammatory cytokines such as IL-1β [56]. Thus, chronic stress drives systemic production of pro-inflammatory cytokines through dysregulation of glucocorticoid receptor sensitivity, and this increase in pro-inflammatory cytokines leads to increased IDO activity. Furthermore, elevated glucocorticoid levels brought about by chronic stress also activate tryptophan-2,3-dioxygenase (TDO) which also promotes the kynurenine pathway of tryptophan metabolism [55]. TDO is largely confined to the liver, but is also expressed in the brain and the gut, making stress and inflammatory pathways highly relevant to the regulation of the kynurenine pathway and serotonin synthesis [55]. 

Activation of the kynurenine pathway produces neuroactive metabolites such as neurotoxic quinolinic acid and neuroprotective kynurenic acid which are generated via kynurenine 3-monooxygenase and kynurenine aminotransferases, respectively. Overstimulation of this pathway under chronic stress has negative implications for the CNS since increased quinolinic acid, particularly in microglia and dendritic cells in the brain, increases inflammation and lipid peroxidation as well as the generation of free radicals by 3-hydroxyanthranilic acid, increasing depression risk [57,58].

Deng et al. [28] and Li et al. [59] showed that tryptophan metabolism in the frontal cortex, hippocampus and gut is affected by CRS and CUMS, respectively. Whilst CUMS produced no changes in colonic serotonin levels, it did result in increased colonic kynurenine, kynurenic acid and both colonic IDO and TDO, suggesting that the kynurenine pathway was favourable in these conditions [59]. Interestingly, a significant positive correlation between colonic kynurenine and hippocampal 3-hydroxykynurenine, the precursor to neurotoxic quinolinic acid, was found, further suggesting that CUMS increased excitotoxicity in the brain via the kynurenine pathway, however while the role of colonic kynurenine is unclear, it is thought to be involved in immunomodulation [55,59]. The production of neurotoxic quinolinic acid or neuroprotective kynurenic acid was region-specific to the colon, cortex and hippocampus, possibly due to differential expression of proinflammatory cytokines and dysfunction of the HPA axis [59]. Proinflammatory cytokines are known to induce IDO as well as the downstream enzyme kynurenine 3-monooxygenase, contributing to quinolinic acid-mediated neurotoxicity [55,59]. Increased cytokines also provide positive feedback to all levels of the HPA axis, increasing glucocorticoid output, thus also increasing TDO expression in the liver and brain, further feeding kynurenine into this pathway [60,61]. In support of these findings, Deng et al. [28] reported increased kynurenine and IDO levels in the hippocampus, frontal cortex and ileum, as well as IDO levels in the colon, in male mice subjected to CRS. TPH 1 and 2, responsible for the conversion of tryptophan to serotonin, were decreased by CRS in the ileum and frontal cortex, respectively, with increased serum levels of kynurenine and 3-hydroxyanthranilic acid, suggesting that, along with inducing depressive-like behaviours, CRS disrupted tryptophan metabolism, particularly favouring the neurotoxic branch of the kynurenine pathway [28]. In support of this, decreased serotonin levels were reported in the frontal cortex and hippocampus of several of the reviewed studies and Angoa-Pérez et al. [62] recently reported that TPH2 knockout male mice exhibited increased depressive-like behaviour [51,63,64,65,66,67,68]. Based on the reviewed studies, inflammation, and dysregulation of the HPA axis as a result of chronic stress are therefore highly relevant to the regulation of serotonin in the brain-to-gut and gut-to-brain connections.

### 3.2. Gut-to-Brain Pathways

#### 3.2.1. Intestinal Barrier Integrity Pathways

Depressive patients are frequently reported to have comorbid inflammatory bowel diseases and stress and depression are known to be associated with low-grade colonic inflammation, possibly due to gut microbiota dysbiosis in association with disrupted intestinal barrier integrity [15,69]. Dysbiosis of the gut microbiota may lead to alterations in the levels of Toll-like receptor (TLR) signalling in the gut, inducing the intestinal inflammatory response [21]. Disrupted intestinal barrier integrity brought about by HPA axis dysfunction may also place the brain and gut at risk since immune signalling and the inflammatory response are also initiated both locally and systemically after the microbiota or lipopolysaccharide endotoxin (LPS) are exposed to intestinal epithelial cells and associated resident immune cells in the lamina propria, inducing TLR-signalling, which activates the NF-κB pathway, subsequently increasing the proinflammatory mediators, inducible nitric oxide synthase and cyclooxygenase-2, ultimately leading to upregulation of proinflammatory cytokines (Figure 5) [21,70]. Moreover, the microbiota or LPS may also enter the circulation (leaky gut) inducing cytokine release through TLR-signalling by dendritic cells and macrophages in the blood stream [21,49]. Since stress and the gut microbiota regulate the permeability of the blood–brain barrier (see the reviews by Kunugi [71] and Segarra et al. [72]), these cytokines may migrate to the brain and activate microglial cells and astrocytes leading to neuroinflammation and alteration in areas of the brain involved in depression [73]. Several studies have reported increased TLR-mediated colonic inflammation in association with increased neuroinflammation across animal models of stress and depression. Lv et al. [52] demonstrated that the number of activated astrocytes and microglial cells in the hippocampus of male rats, who received faecal microbiota transplantation from depressed rats, compared to control rats was increased based on glial fibrillary acidic protein—(GFAP/Iba-1) positive cells in addition to increased necrosis and shedding of the intestinal epithelium. CSDS exposure induced hippocampal microglial activation together with increases in proinflammatory cytokines IL-6, TNF-α, IL-1β in both the hippocampus and colon [54]. In female mice, CUMS activated hippocampal glial cells (evidenced by increased IL-1β and NLRP3 inflammasome) in association with increased colonic IL-1β [43]. In CRS-induced depressive male mice, the inflammatory cytokines IL-6, TNF-α, IL-1β and IL-22 were increased in colon tissues, with increased serum LPS, IL-6 and TNF-α and increased TNF-α levels in the hippocampus [69]. Contrarily, in a different study, CRS was reported to have no significant impact on cytokine levels in plasma, cecum tissue or colon tissue extracts [74]. The contrasting results in the CRS studies may be related to the duration of the CRS protocol since Stenman et al. [74] utilized a 2-week protocol in comparison with Xiao et al. [69] who utilized a 4-week protocol. MS showed similar effects to 4 weeks of CRS since it activated expression of TLR4, TNF-α and IL-1β in colon tissue and activation of the neuro-immune response was evident from increased TLR4 and IL-1β, but not TNF-α in the hippocampus in male MS mice compared to controls [23]. Early life stress, which the MS model achieves, is known to be robust in inducing neurobiological changes via dysregulation of the HPA axis and immunological alterations related to depression due to the vulnerability of the brain and immune system during youth [2,6].

Recently, it was demonstrated that pubertal LPS administration in male and female mice produced anxiety-like behaviours in males and depressive-like behaviours in females, that were sustained into adulthood, along with sex differences in central and peripheral inflammatory cytokines, suggesting LPS exposure alters different neural pathways in each sex [75].

Thus, colonic inflammation in association with neuroinflammation may only be achieved in those animal models of stress and depression that utilize protocols with a longer duration or which target the brain during periods of vulnerability. Since HPA axis dysfunction has been associated with irritable bowel comorbidities in depression, these inflammatory changes in the gut and the brain may be linked with intestinal barrier integrity [15].

Impairments of the intestinal barrier may therefore act as a gateway to these TLR-mediated immune responses in the brain. Characterized by impaired expression of tight junction proteins in the intestinal mucosa, compromised intestinal barrier integrity allows for bacteria or LPS to enter the circulation initiating an inflammatory response and increasing circulating proinflammatory cytokines and depressive-like behaviour [73]. This has been supported by findings of increased serum LPS in depressive patients [13,69]. In the reviewed studies, stress had a clear effect on intestinal barrier integrity since tight junction proteins were decreased in stressed mice exposed to CUMS, CRS and CSDS [21,69,77,78]. According to Yamagishi et al. [78], these effects are a result of gut microbial dysbiosis as some bacteria are known to modulate tight junction expression directly. Furthermore, increased inflammation driven by perturbations in other pathways such as efferent vagal nerve signalling is also known to weaken tight junctions in the gut [49]. In the reviewed studies, animals fed non-purified standard diets were also consistently shown to have increased histology scores of colonic epithelial cells, showing increased necrosis, shedding and infiltration of inflammatory cells, across the CUMS, CRS and MS stress models, suggesting that chronic stress has deleterious effects on general gut health, regardless of the modality of the stressor or variations in non-purified standard diets [23,28,52,69].

Another critical link between stress and intestinal barrier function involves host production of antimicrobial peptides by cells such as Paneth cells, which reside in intestinal crypts where they provide trophic support for epithelial cells and secrete immunomodulating and antimicrobial peptides, such as α- and β-defensin, to maintain intestinal homeostasis [79]. β-defensin expression is known to be regulated by MUC2, a major component of the mucus layer in the gut which serves as an additional barrier between gut microbes and epithelial cells [29]. The mucus layer is also composed of a loose outer layer colonized by the gut microbiota and a firmer inner layer which serves as a diffusion barrier, separating the microbiota from epithelial cells [80]. The study by Gao et al. [29] showed that 4 weeks of CRS was sufficient to decrease colonic MUC2 expression and goblet cell numbers. Paneth cells have also been demonstrated to have a close association with MUC2 in that its presence around Paneth cells facilitates the secretion of antimicrobial peptides into the mucus layer [81]. Suzuki et al. [76] showed that CSDS had deleterious effects on the number of Paneth cells and their secretory granules in the small intestine. 

Short chain fatty acids (SCFA), bacterial metabolites produced via the digestion of dietary fibre, may too play a role in the regulation of intestinal barrier integrity as well as behaviour [11]. Through facilitation of tight junction assembly, SCFAs may also regulate blood–brain barrier integrity [71]. Disruptions to intestinal barrier integrity are often associated with lowered SCFA levels and may inhibit production and release of serotonin by enteroendocrine cells [52]. Consequently, neuroinflammation is increased and the bidirectional communication between gut and brain is impaired, increasing the incidence of depression [82]. In support of this, decreased faecal SCFA levels have also been reported in depressive patients and animal models of stress and depression [54,57,83]. This is of relevance to the gut–brain axis since, according to Caspani et al. [57] and Egerton et al. [83], the SCFA butyrate, may directly affect permeability of the intestinal wall by increasing the expression of tight junctions (occludin and ZO-1). Another possible mechanism behind the correlations between SCFAs and behaviour may occur in the gut and involve G-protein coupled receptor 43, otherwise known as free fatty acid receptor 2. Free fatty acid receptor 2 is expressed on enteroendocrine cells and is known to bind SCFAs [11,57]. Importantly, free fatty acid receptor activation stimulates the release of glucagon-like peptide-1 and peptide YY into the blood stream, which regulate post-prandial release of insulin and satiety, respectively [84]. Chronic stress is also known to decrease GLP-1 levels in the paraventricular nucleus of the hypothalamus, dysregulating eating behaviour [85]. Amongst animal models of depression, Oh et al. [21] previously found CUMS decreased colonic expression of free fatty acid receptor 2 and Tian et al. [77] showed free fatty acid receptor 2 and 3 expression to be decreased by CSDS, suggesting that stress can modulate the gut’s capacity to respond to SCFA, in turn modulating eating behaviour.

Naturally, the gut microbiota produce many different metabolites other than SCFAs, many of which are neuroactive and are associated with some of the gut-to-brain pathways covered in this review such as amino acid metabolites and lactate (see the reviews by Ortega et al. [86] and Tran and Mohajeri [87]). Of note are the metabolites of tryptophan including kynurenine metabolites, indole and indole acid derivatives which bind aryl hydrocarbon receptors thus modulating the immune response [86,88]. Activation of aryl hydrocarbon receptors by tryptophan metabolites also stimulates IDO expression, further stimulating the neurotoxic branch of the kynurenine pathway and potentially increasing depression risk [88,89].

Evidently, chronic stress can modulate the permeability of the intestinal barrier by diminishing the mucus layer, decreasing tight junction protein expression and reducing the capacity of the intestinal barrier’s response to SCFAs. A diminished mucus layer in the gut may also contribute to dysbiosis of the gut microbiota, leading to decreased production of SCFAs, increasing colonic inflammation and further impacting tight junction protein expression and increasing intestinal barrier permeability, ultimately exposing the circulation to the bacterial endotoxin, LPS, or other bacterial metabolites or ligands. TLR-mediated immune responses increase circulating cytokines which can cross the blood–brain barrier, thus increasing neuroinflammation and inducing depressive-like behaviours. Thus, with dysbiosis resulting in profound effects on the gut–brain axis, specific alterations to the microbiota and how these impact the gut–brain axis are of importance.

#### 3.2.2. Gut Microbiota Pathways

Notably, the murine and rodent gut microbiota are not necessarily comparable to the human gut microbiota, since it is known that a considerable proportion of the human gut microbiota are unable to colonize the guts of such animals due to the absence of factors such as human genotype and lifestyle [90]. Despite that, preclinical animal models have been crucial in elucidating potential pathways of the gut–brain axis. Moreover, despite translational uncertainty to humans, it remains plausible that similar findings may arise in well-phenotyped depressive human populations [91].

In both human and animal studies of stress and depression, the types of gut microbes reported in the literature are typically referred to on a particular level of the biological order of classification, ranging from less specific levels, such as the phylum to more specific levels such as the genus [5]. At each level, the microbial diversity can be reported using separate measures known as α-diversity and β -diversity [5,92]. Bridgewater et al. [93] demonstrated that the gut microbiota diversity and structure differ markedly in male and female mice, highlighting the existence of sex differences in the gut microbiota, however most studies report stress-induced alterations in the gut microbiota in male animals. Thus, the sections to follow will have a broader focus on male features and any female specific changes will be highlighted.

##### Gut Microbial Diversity

Microbial α-diversity is a measure of the abundance and uniformity of species within the microbial community of a sample. Of the reviewed studies which reported α-diversity, most studies used indices such as Shannon and Chao1, and others used Simpson and ACE indices. Of these, Kai et al. [94] and Xiao et al. [69] reported significant increased diversity, but most studies reported significant decreased diversity at the phylum level in stressed animals [21,42,51,52,54,63,64,68,92,95,96,97,98]. Some studies reported no significant changes to α-diversity at the phylum level. At the genus level, only Guo et al. [99] reported a significant increase in α-diversity and only Szyszkowicz et al. [100] reported α-diversity at the species level, however this did not reach significance.

β -diversity is a measure which compares the compositions of microbial communities between samples [92]. β-diversity was reported across the reviewed studies using Principle Coordinate Analysis or Principle Component Analysis with Adonis or Bray–Curtis analyses. Several recent studies have reported significant differences in β-diversity at the phylum level [21,42,51,52,62,64,68,92,94,95]. However, one study reported a significant difference at the genus level in stressed animals versus controls and a separate study reported significant differences at both the phylum and genus levels [83,101]. Seven studies reported no significant differences to β-diversity. Thus, this indicates no apparent dependable changes to the diversity of the gut microbiota in animal models of stress and depression. A review by Barandouzi et al. [102] similarly found no reliable changes in both α- and β-diversity in human studies.

Despite this, there is considerable evidence for the involvement of the gut microbiota in stress and depression. For example, Lv et al. [52] and Han et al. [96] demonstrated that transplantation of faecal samples from animals exhibiting depressive-like behaviours into germ-free counterparts induced the same behaviours in the germ-free counterparts. Transplantation of faecal samples from healthy control animals does not induce the same effects [96]. Thus, these effects may be associated with specific microbial composition rather than overall perturbations in diversity, although it is equally as likely that these effects are related rather to the transfer of other components present in the faecal samples such as metabolites and hormones, but are nonetheless related to the stress response.

##### Gut Microbial Composition

Considering the composition of gut microbiota in the reviewed studies, at the phylum level, the most frequently reported bacterial phyla include Firmicutes, Bacteroidetes, Actinobacteria and Proteobacteria, with the two most dominant phyla being Firmicutes and Bacteroidetes [51,52,100,103] (Table 1). The ratio of Firmicutes to Bacteroidetes has previously been demonstrated to be decreased in depressed patients [14]. In the animal studies of stress and depression covered in this review, most studies reported that stress increased Firmicutes abundance, however Bacteroidetes abundance was more variable (Table 1). Interestingly, Duan et al. [104] showed Firmicutes abundance was higher, and Bacteroidetes abundance was lower in mice which did not respond to Escitalopram anti-depressant treatment than those that did respond to treatment. Given that the causes of treatment resistance in up to 30% of depressive patients remain elusive, future studies of gut microbiota disturbances in non-responders are warranted [105]. Actinobacteria and Proteobacteria have been shown to be altered in depressive patients compared to healthy individuals [5,14], however in animal models of stress and depression alterations in these two phyla were inconsistent (Table 1).

At the genus level, changes to the compositions of the gut microbiota were less consistent than those seen at the phylum level, but more widely studied across the different models. Over 200 different genera make up the Firmicutes phylum, however despite this, 95% of this dominant phylum is made up of *Clostridium* [118]. The study by Chen et al. [119] showed a decrease in *Clostridium* abundance consistent with a decrease in Firmicutes, however the remainder of the studies reviewed did not demonstrate this. Oh et al. [21] showed a non-significant increase in *Clostridium* associated with an increase in Firmicutes contrary to depressive patients having decreased levels of Firmicutes [57,120]. Consistent with increased Bacteroidetes abundance in depression, *Bacteroides* has also been shown to be increased in animal models of depression [99,121]. A CUMS-induced decrease in Bacteroidetes abundance was also reported in female mice [43].

Of all the most commonly reported genera, *Turicibacter* and *Allobaculum*, both belonging to Firmicutes, were the only genera to be consistently decreased in stressed animals across the different models (Table 2). Fung et al. [122] previously demonstrated that colonization of the colon by *Turicibacter sanguinis* is modulated by gut-derived serotonin levels, since *T. sanguinis* is capable of utilizing colonic serotonin due to a serotonin-sodium symporter homologous to the serotonin transporter found on intestinal epithelial cells. While it has been established that gut-derived serotonin is likely separate to CNS levels due to the poor ability of serotonin to cross the blood–brain barrier, it has been proposed that it may have an indirect effect on the brain possibly through the activation of serotonin receptors on afferent vagal neurons in the intestinal mucosa [123]. Enterochromaffin cells release serotonin both luminally and basally, thus, the decreased levels of *Turicibacter* reported in the reviewed studies may be associated with decreased colonic serotonin levels that arise from stress-induced alterations in enterochromaffin cell function, which may also have negative implications for vagal nerve signalling [124]. Regarding *Allobaculum*, Suzuki et al. [76] reported that CSDS reduced faecal α-defensin levels, and this was positively correlated with *Allobaculum* abundance, indicating that CSDS disturbed *Allobaculum* abundance via perturbations in intestinal function of Paneth cells and likely also decreased the mucus layer of the intestinal barrier due to the close association of Paneth cells and the regulation of this important component. CUMS-induced alterations in *Allobaculum* abundance have been reported in female C57BL6 mice, however these were inconsistent across studies likely due to differing durations in the CUMS protocol [43,93]. Thus, stress may modulate the gut microbiota through modifications in the luminal environment of the gut driven by alterations in enteroendocrine function and physical changes to the intestinal mucus barrier which the gut microbiota colonise. This increases the permeability of the gut and increases the systemic TLR-mediated immune response to bacterial ligands resulting in neuroinflammation [73].

Overall, considering Table 1 and Table 2, the apparent variability in the alterations to the gut microbiota in animal models of depression may be due to differences in methodology across the studies. The different stress paradigms established to model depression in rodents are known to emulate different characteristics of depression, but do not all produce the same neurobiological or behavioural outcomes [6]. The selected site of sample collection may also contribute to the variability seen in the composition of the gut microbiota in animal models of depression (Table 1 and Table 2). It is known that the distribution of the gut microbiota varies along the gastrointestinal tract, and it was previously shown in humans that the quantity of facultative anaerobes can differ significantly in caecal versus faecal samples [129,130,131].

One of the most overlooked factors that may affect the variability of gut microbiota data across recent studies is the selection of standard diets. Toyoda et al. [132] previously showed differences, through chronic feeding alone, in the abundance of specific genera of up to 40% in mice fed non-purified versus purified standard diets. This demonstrates that compositional gut microbiota changes are highly influenced by diet. In a previous study, Toyoda et al. [132] showed mice of the same strain were less susceptible to social defeat stress when fed a non-purified versus a purified diet, highlighting the need for the appropriate selection of feed for the reproducibility of neuroscience research [133]. Thus, the role of diet in the gut–brain axis and behaviour needs to be evaluated in conjunction with the changes in gut microbial composition.

## 4. The Effects of Diets on the “Gut–Brain” Pathways across Animal Models of Stress and Depression: Implications for Behaviour

It is well known that diet is a key modulator of gut microbial composition. In humans, good quality plant-based diets such as the Mediterranean diet have been shown to reduce pathogenic bacteria in the gut, increase *Bifidobacterium* and *Clostridium*, as well as lower the risk of depression, while poorer quality diets such as the Western diet have been shown to reduce *Lactobacillus* in the gut, reduce overall gut microbial diversity and have been associated with increased depression risk [14,15]. Evaluating the effects of diets on the brain-to-gut and gut-to-brain mechanisms in animal models of stress and depression may aid in the elucidation of the pathophysiology of depression and may provide novel therapeutic approaches. Thus, we will review the effects of probiotic and plant-based prebiotic supplementation in animal models of stress and depression, particularly in studies which evaluated relevant alterations discussed earlier in this review in both the gut and the brain to better appreciate the bidirectional lines of communication involved. To our knowledge, studies under this scope which utilized diets or supplements mimicking a westernised diet are lacking, thus we will conclude with a brief summary of key findings of interest for such diets.

### 4.1. Prebiotics and Probiotics Affect Neuroplasticty along the Gut–brain Axis

According to Bear et al. [15], healthier diets rich in fruit and vegetables typically contain higher amounts of prebiotics such as polyphenols and prebiotic soluble fibres including fructooligosaccharides and galactooligosaccharides and are associated with decreased risk of depression. Widely studied for their anti-inflammatory and antioxidant effects, polyphenols are known to modulate the gut microbiota and, as a dietary compound, are known to be largely indigestible without biotransformation by the colonic microbiota [134]. Observational studies in humans have shown an inverse association between polyphenol intake and depression risk [135]. Such effects may be due to increased *Bifidobacterium* and *Lactobacillus* abundance since polyphenols are known to modulate the gut microbiota by increasing these bacterial strains [136]. In animal models of stress and depression, supplementation with *Bifidobacterium* and *Lactobacillus* appeared to have an effect on colonic serotonin in mice, and possibly tryptophan availability, since it increased colonic TPH1 and 5-HTP levels, previously decreased by CUMS [21,42]. Tian et al. [54] did not report on colonic serotonin levels, but showed *Bifidobacterium longum* supplementation increased levels of serum 5-HTP, consistent with increased serum serotonin levels found by Oh et al. [21] following *L. rhamnosus* supplementation, suggesting downregulation of the kynurenine pathway in the colon and periphery. Interestingly, in the reviewed studies which used rats in their animal models of stress and depression, probiotic supplementation had the opposite effect on colonic serotonin and TPH1 levels compared with the studies in mice, suggesting that colonic tryptophan metabolism in rats may be different to mice (Table 3). Indeed, according to Murakami and Saito [137], in rats, responses to immune activation, highly relevant to the kynurenine pathway of tryptophan metabolism, do not mimic those of humans and LPS administration in rats also does not affect levels of quinolinic acid in the brain, suggesting that tryptophan metabolism along the gut–brain axis is different in rats compared to mice. 

Nonetheless probiotic and prebiotic supplementation appear to have consistent effects on the brain and behaviour. In the studies reviewed, dietary supplementation with fructooligosaccharides and galactooligosaccharides, *Bifidobacteria* or *Lactobacilli* attenuated depressive- and anxiety-like behaviours in rats and mice [21,42,51,54,75,97,98,115]. Oh et al. [21] and Tian et al. [42] reported that supplementation with these probiotics increased hippocampal levels of the serotonin receptor, 5HT1a. Increased serotonin levels in the frontal cortex were also consistently reported as a result of probiotic or prebiotic supplementation [51,54]. Jasmine tea, considered a prebiotic, has also been demonstrated to restore frontal cortex and hippocampal serotonin in a similar manner to fluoxetine in association with restoration of the Firmicutes:Bacteroidetes ratio [68]. Furthermore, supplementation with *B. longum*, *B. breve* and *Clostridium butyricum* all reversed decreases in hippocampal and frontal cortex BDNF expression caused by CUMS. *Lactobacillus* probiotics were also able to restore BDNF levels in the frontal cortex in MS and the hippocampus in CRS [66,98]. Therefore, the effects of prebiotics such as fructooligosaccharides and galactooligosaccharides, or probiotics such as *Lactobacillus* or *Bifidobacterium*, on behaviour in these animal studies may be a result of the restoration of neuroplasticity brought about by increased serotonin and BDNF levels in key brain regions involved in depression pathophysiology, however colonic serotonin modulation by the gut microbiota appears to be host species-dependent and separate to the CNS, while the mechanism behind the modulation of CNS BDNF levels by the gut microbiota remains to be elucidated.

### 4.2. Diet, Stress and Intestinal Barrier Integrity

Probiotics such as *Bifidobacteria* and/or *Lactobacilli* may also reduce dysregulation of the HPA axis brought about by chronic stress, and in turn regulate gut health. Changes in the composition of the microbiota have been attributed to overactivation of the HPA axis and overproduction of corticosterone [23], leading to increased intestinal permeability [138]. Gut permeability is also upregulated by a higher Firmicutes:Bacteroidetes ratio [139] and downregulated by *Bifidobacteria* [140]. In CUMS, increased hypothalamic CRF and serum corticosterone levels were attenuated by *Bifidobacterium* and *Lactobacillus* supplementation [21,42,54,66]. *Bifidobacterium* supplementation was also demonstrated to reverse a CUMS-induced increase in circulating TNF-α, possibly due to a reduction in LPS or microbiota exposure to the circulation, although the mechanism of LPS reduction is unclear [42]. A possible explanation may be alterations in the expression of TLRs at the level of the paraventricular nucleus of the hypothalamus. Murray et al. [75] showed that administration of the probiotic kefir to mice exposed to LPS reduced TLR4 expression in the paraventricular nucleus in association with reduced anxiety-like behaviour and stress activity in male mice, however probiotic kefir only reduced depressive-like behaviour in females. These effects resulted in reduced central and peripheral inflammatory cytokines in a sex-specific manner [75]. These restorative effects on the HPA axis and inflammation may therefore have similar effects on gut health due to the close association between stress and poor gut health.

While stress had a clear effect on intestinal barrier integrity since tight junction proteins were decreased in stressed mice exposed to CUMS, CRS and CSDS [21,54,66,69,78], the studies by Oh et al. [21] and Tian et al. [77] showed that decreased colonic barrier integrity brought about by stress could be abolished by supplementation with *Lactobacillus* and *Clostridium* probiotics, respectively, demonstrating that intestinal permeability may not only be under regulation by stress. Wang et al. [66] showed similar effects for the jejunum and ileum with *Lactobacillus* in CRS-treated mice. These effects found on intestinal barrier integrity may involve anti-inflammatory mechanisms since the study by Oh et al. [21] showed that increased intestinal barrier integrity brought about by *Lactobacillus* was accompanied by colonic and neuronal suppression of both the pro-inflammatory mediators, inducible nitric oxide synthase and cyclooxygenase-2. Wang et al. [66] demonstrated that *Lactobacillus* could attenuate CRS-induced increases in ileal TNF-α and IFN-γ and increase the anti-inflammatory cytokine IL-10. Similarly, *Clostridium* supplementation attenuated CSDS-induced increases in colonic inflammation and hippocampal microglial activation. 

Since the integrity of the intestinal barrier is associated with SCFA levels, probiotics may strengthen intestinal barrier activity via SCFA production in the gut. Based on the studies reviewed, SCFA production may be species-dependent since supplementation with *B. breve*, unlike *B. longum*, restored colonic SCFA levels previously decreased by CUMS [42]. However, probiotic supplementation with *Lactobacillus* or *Clostridium* was shown to restore expression of colonic free fatty acid receptors, previously decreased by chronic stress, suggesting restoration of the colon’s capacity to respond to SCFAs [21,77]. Overall, these effects may have strengthened the intestinal barrier, preventing exposure of the circulation to bacterial ligands such as LPS, thus preventing TLR-mediated neuroinflammation and depressive-like behaviour. Indeed, Burokas et al. [115] showed in their study that general increases in SCFA levels brought about by fructooligosaccharide and galactooligosaccharide supplementation were strongly correlated with positive changes in anhedonic, depression-like and social behaviour. Prebiotics such as guar gum may also beneficially shift the SCFA profile of the gut which has been shown to alleviate depressive-like behaviour by maintaining striatal and hippocampal serotonergic and dopaminergic neurotransmission; however, how SCFAs mediate these effects in the brain requires further investigation [120].

Contrary to the positive effects of probiotics and plant-based prebiotics on the gut–brain axis, the effects of diets which resemble a westernised diet such as high-fat diets or cafeteria diets on the pathways discussed remain to be elucidated. Current literature shows such westernised diets impair memory, alter neuroplasticity in key brain regions involved in depression, and increase neuroinflammation [141,142,143]. However, whether these effects modulate stress-related brain-to-gut and gut-to-brain mechanisms is unclear since studies evaluating brain and behaviour in conjunction with gut health in well-established animal models of stress and depression are lacking. de Sousa Rodrigues et al. [144] showed that stressed mice fed a high fat, high fructose-diet had increased anhedonia and anxiety-like behaviour in association with increased gut permeability and circulating TNF-α and IL-6 levels compared to their non-stressed counterparts. Yet, hippocampal cytokine expression was unaffected by both diet and stress, however this may have been related to the predatory stress model used [144]. de Sousa Rodrigues et al. [144] suggested that the increased gut permeability may be driven by high fat intake activating intestinal mast cells, leading to increased inflammatory cytokine production which may have promoted gut microbial dysbiosis, although the gut microbiota was not assessed in their study. Nonetheless, according to a review by Rohr et al. [145], high-fat diets increase intestinal permeability and local TLR-mediated inflammation, diminish the mucus layer of the gut and alter the composition of the gut microbiota. Further studies have shown that rodents fed high fat or westernised diets have decreased gut microbial diversity, suggesting that such diets may indeed have deleterious effects on the bidirectional mechanisms between the gut and the brain [93,146,147]. Interestingly, the composition of the gut microbiota in unstressed female mice fed a high fat diet has been shown to resemble that of female mice exposed to CUMS, but fed a standard diet; however, the same effect is not observed in males, thus females may be more susceptible to the potential negative effects of high fat diets on the gut–brain axis [93]. Hatton-Jones et al. [111] showed a Western diet decreased colonic tight junction protein expression in male mice, but this did not result in increased depressive-like behaviour and these effects were not synergistic with those of CRS. Yet, Ranyah Shaker et al. [148] demonstrated male Wistar rats fed a high-fat diet had increased dopamine and glutamate, but decreased serotonin in the brain, with increased circulating IL-6 in association with decreased *Clostridium* and *Bacteroides* abundance. In support of this, Saiyasit et al. [149] showed that Wistar rats fed a high-fat diet exhibited gut microbial dysbiosis after 2 weeks, which was followed by decreased hippocampal plasticity and dendritic spine density in association with cognitive decline, suggesting dysbiosis may drive brain pathology. Thus, westernised diets in rodents appear to have negative consequences for brain health possibly driven by gut-to-brain pathways, however the relevance of these changes in the context of stress and depression are unclear, warranting further study of westernised diets in common animal models of stress and depression.

## 5. Conclusions and Future Directions

The emerging role of the gut–brain axis in anxiety and depression emphasises the importance of understanding the bidirectional interaction associated with the composition of the gut microbiota and the biological pathways through which diet and stress affect brain health in a sex-specific manner. 

Neuroplasticity and immune activation are coordinated by multiple systems including the CNS and neuroendocrine system. The pathways involved including inflammation, the gut microbiota, tryptophan-kynurenine metabolism, and the HPA axis identified in this review, arise mostly from animal models. A summary of the main conclusions drawn in this review is presented in Box 1. To further delineate the complex and interacting modes of action, studies in humans diagnosed with clinical depression are needed. Furthermore, some of these pathways require further elucidation in animal models, such as the role of BDNF changes in the gut during chronic stress. Of note is the importance of the microbiota composition, where specific species of bacteria within the same family play different roles associated with either pro-inflammatory or anti-inflammatory properties and regulation of colonic tryptophan availability, with downstream effects on neuroplasticity and behaviour.

Box 1.A summary of the effects of stress, diets and sex on the bidirectional lines of communication between the brain and the gut in animal models of stress and depression. HPA—hypothalamic-pituitary-adrenal axis; ANS—autonomic nervous system; LPS—lipopolysaccharide; TLR4—toll-like receptor 4.
**Effects of stress**
Activation of HPA axis and ANSInhibition of afferent vagal nerve fibresIncreased intestinal, systemic inflammation and neuro-inflammationDysregulation of tryptophan metabolism by upregulation of the kynurenine pathway in the brain, liver and gutImpairment of intestinal barrier integrity and reduction of mucus layer of the gutGut microbial dysbiosis

**Effects of diets**
Prebiotics and probiotics show promise as dietary supplements which may alleviate depression pathophysiology*Bifidobacteria* and/or *Lactobacilli* supplementation restore neuroplasticity in key brain regions involved in depression*Bifidobacteria* and/or *Lactobacilli* supplementation restore intestinal barrier integrity and negative feedback in the HPA axisWesternised diets impair neuroplasticity in key brain regions involved in depressionWesternised diets impair intestinal barrier integrity and the mucus layer of the gut

**Effects of sex**
Male bias exists in studies investigating the effects of stress on the multi-directional communication between the brain, the gut and gut microbiotaThe diversity and composition of the gut microbiota differ by sex, although further studies need to establish stress-induced alterations in femalesProbiotics reduce LPS-induced depressive-like behaviour in females and anxiety-like behaviour and stress activity associated with increased TLR4 expression in malesFemales may be more susceptible to the negative effects of high fat diets on the gut-brain axis


Going forward, consideration should also be given to other constituents of the gut microbiota other than bacteria, such as archaea and fungi, and whether these contribute to stress and depression pathophysiology or whether stress affects their abundance. Additionally, the current literature reviewed in this study was gender biased towards evaluating the gut–brain axis in male animals, while it is well known that depression incidence in humans is higher in females. Given the sex-specific differences in the gut–brain axis highlighted in this review, future studies are encouraged to include female mice while evaluating the gut–brain axis to improve understanding of gender differences of stress and depression. Finally, further studies are required to elucidate the effects of westernised diets on the multiple pathways involved in the bidirectional communication between the brain and gut in animal models of stress and depression.

In conclusion, the role of dietary interventions and response to treatment will enhance our understanding of depression pathophysiology and the interplay between diet, gut microbiota and depression. The ability to understand gut dysbiosis and the levels of biomarkers related to these pathways in individuals will lead to the emergence of knowing what works for individuals, and potentially enable the design of new and targeted strategies for depression and improved mental health, such as the potential combination of anti-inflammatory and antidepressant therapy.

## Figures and Tables

**Figure 1 ijms-23-02013-f001:**
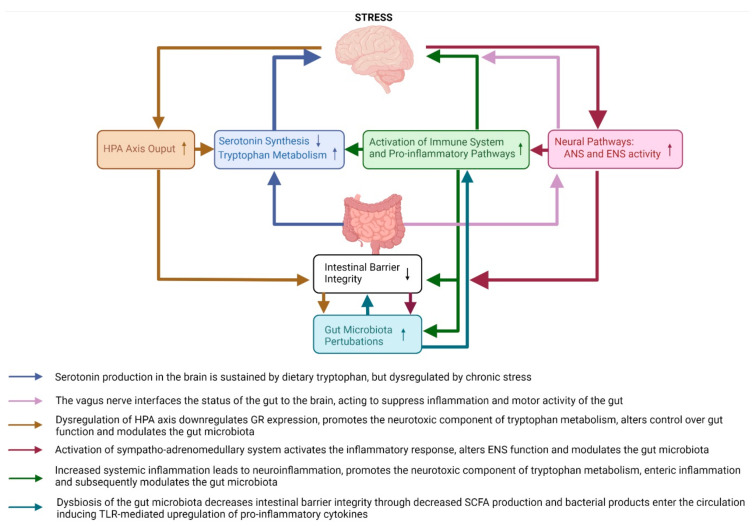
An overview of the major brain-to-gut and gut-to-brain pathways affected by chronic stress in animal models of stress and depression. The pathways affected by chronic stress include the HPA axis, neural pathways, immune system pathways, serotonin and tryptophan pathways, intestinal barrier integrity and gut microbiota pathways. HPA—hypothalamic–pituitary–adrenal axis; ANS—autonomic nervous system; ENS—enteric nervous system; GR—glucocorticoid receptor; SCFA—short chain fatty acids; TLR—toll-like receptor.

**Figure 2 ijms-23-02013-f002:**
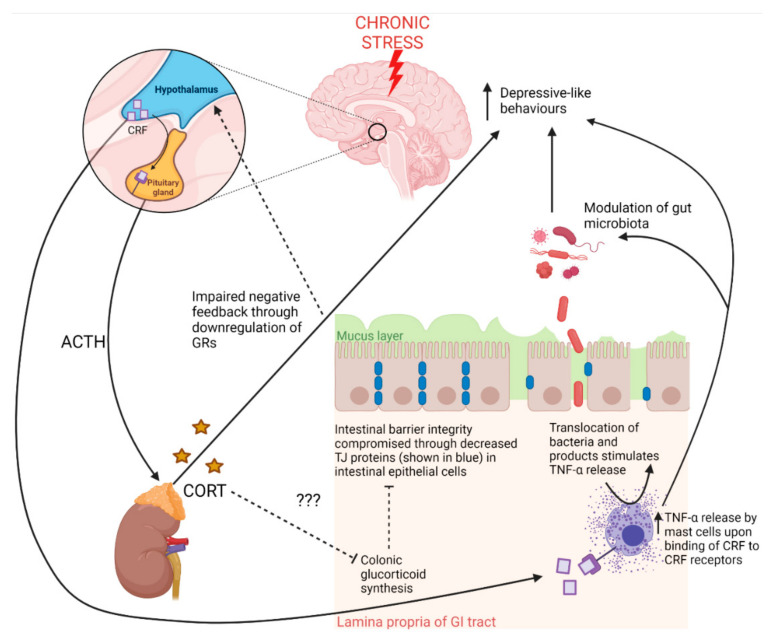
The contributions of the hypothalamic–pituitary–adrenal (HPA) axis to gut pathology and behaviour in response to chronic stress. Exposure to stress stimulates corticotropin releasing factor (CRF) release from the hypothalamus, which induces pituitary release of adrenocorticotropic hormone (ACTH) [2]. ACTH stimulates the release of corticosterone (CORT) from the adrenal cortex [2]. Chronic stress downregulates hippocampal glucocorticoid receptors (GRs), which bind CORT regulating negative feedback of the HPA axis under normal conditions [6]. Chronic stress thus leads to chronically elevated CORT levels, which are known to lead to depressive-like behaviours. Under chronic stress, CRF production by the hypothalamus may also reach the circulation where CRF may travel to the gut and bind CRF receptors on mast cells, inducing intestinal inflammation and compromising intestinal barrier integrity [32,34]. Some evidence indicates that colonic glucocorticoids may contribute to the maintenance of intestinal barrier integrity and that colonic glucocorticoid synthesis occurs through separate mechanisms to adrenal-derived glucocorticoids [34]. Adrenal-derived glucocorticoids may also inhibit colonic glucocorticoid synthesis, thus corroborating findings of impaired intestinal barrier integrity existing with a dysregulated HPA axis [34]. GI tract—gastrointestinal tract; TJ—tight junction proteins.

**Figure 3 ijms-23-02013-f003:**
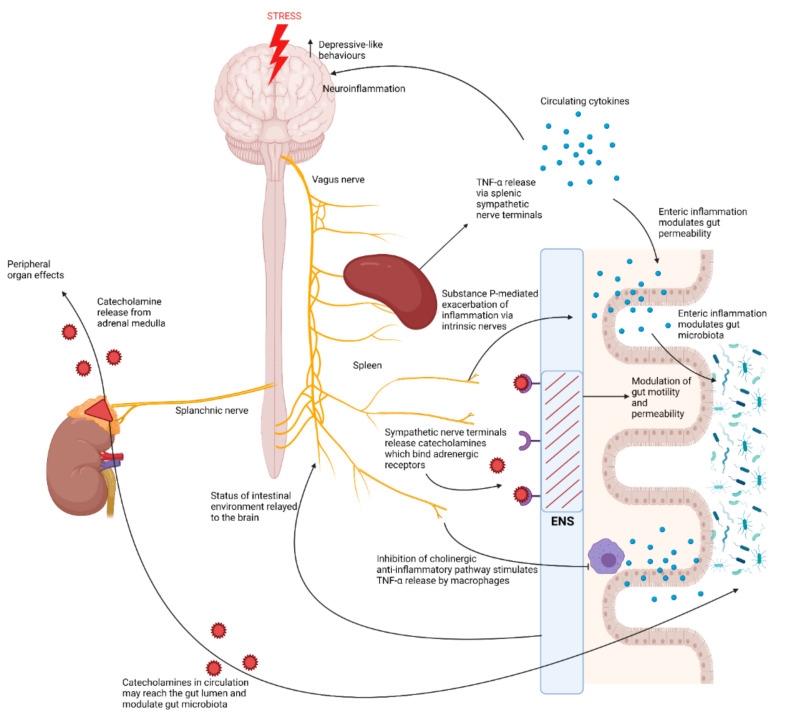
The brain-to-gut effects of activation of the autonomic nervous system in response to chronic stress. Exposure to stress activates the sympatho–adrenomedullary system which results in the release of catecholamines from the adrenal medulla via activation of the splanchnic nerve [10]. Catecholamines released by the adrenal medulla exert various effects on different peripheral organs, however they may also travel via the circulation to the gut where they may diffuse into the lumen and modulate the gut microbiota [10]. Exposure to stress also exerts effects over the enteric nervous system (ENS), by driving activation of sympathetic nerve terminals which release catecholamines that bind adrenergic receptors in the ENS, thus modulating intestinal motility and permeability [10]. Chronic stress also inhibits the vagus nerve. This results in increased splenic TNF-α release, exacerbated by Substance-P-driven responses by intrinsic nerves of the ENS [39]. Inhibition of the vagus nerve by chronic stress also inhibits the cholinergic anti-inflammatory pathway, stimulating TNF-α release by macrophages in the gut [12]. This increase in pro-inflammatory cytokines may lead to intestinal inflammation, compromising intestinal barrier integrity, but may also modulate the gut microbiota [11,39].

**Figure 4 ijms-23-02013-f004:**
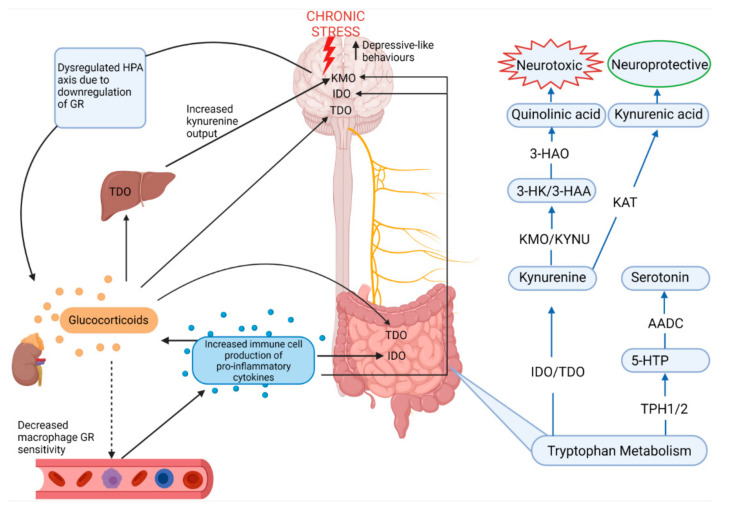
Stress and the regulation of the major metabolic pathway of tryptophan: the kynurenine pathway. From the bottom right, in a basal state 5% of dietary tryptophan is metabolized into serotonin while 95% of tryptophan is metabolized into kynurenine [28,55]. Chronic stress is known to increase pro-inflammatory cytokines and elevate glucocorticoid levels through dysregulation of the HPA axis [6]. Glucocorticoids enhance TDO expression in the brain, liver and gut, while pro-inflammatory cytokines enhance IDO expression in the brain and gut as well as KMO expression in the brain [28,55]. Since KMO is the first enzyme involved in metabolizing kynurenine into neurotoxic quinolinic acid and since kynurenine is capable of crossing the blood-brain-barrier, chronic stress therefore increases neurotoxicity in the brain by upregulation of the neurotoxic branch of the kynurenine pathway [55,58]. HPA—hypothalamic–pituitary–adrenal axis; GR—glucocorticoid receptor; TDO—tryptophan-2,3-dioxygenase; IDO—indolamine-2,3-dioxygenase; KMO—kynurenine 3-monooxygenase; KYNU—kynureninase; 3-HK—3-hydroxykynurenine; 3-HAA—3-hydroxyanthranilic acid; 3-HAO—3-hydroxyanthranilic acid oxidase; KAT—kynurenine aminotransferases; 5-HTP—5-hydroxytryptophan; TPH1/2—tryptophan hydroxylase; AADC—aromatic amino acid decarboxylase.

**Figure 5 ijms-23-02013-f005:**
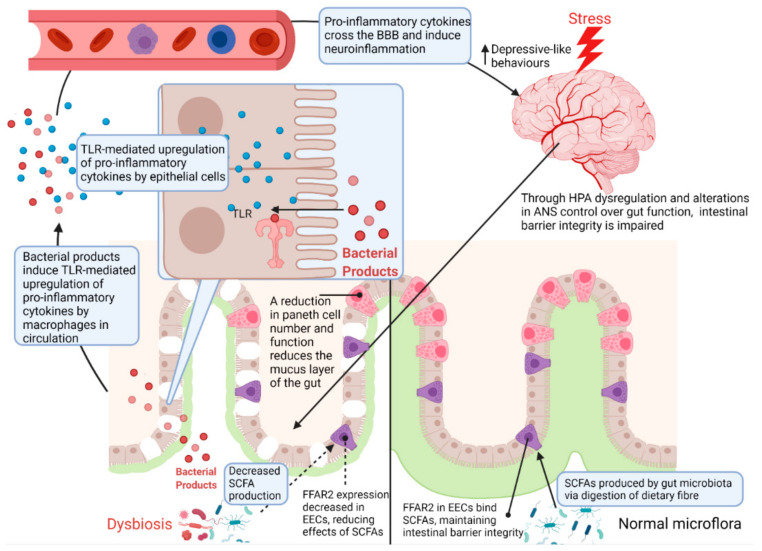
Regulation of the intestinal barrier by chronic stress. Chronic exposure to stress results in dysregulation of the HPA axis and ANS control over gut function [2,6]. As a result, Paneth cell number and function is reduced, contributing to microbial dysbiosis through a reduction in the mucus layer of the gut [29,76]. This reduction of the mucus layer also allows bacteria and their associated products to reach the gut epithelium [49]. Since dysregulation of the HPA axis results in a decrease in tight junction proteins, bacterial products such as LPS can reach epithelium, the lamina propria and circulation where a TLR-mediated pro-inflammatory response is initiated by macrophages [49]. These pro-inflammatory cytokines ultimately travel to the brain where they can cross the blood–brain barrier and increase neuroinflammation, eliciting depressive-like behaviours [2,49]. HPA—hypothalamic–pituitary–adrenal axis; ANS—autonomic nervous system; SCFA—short chain fatty acids; FFAR2—free fatty acid receptor 2; EEC—enteroendocrine cell; TLR—Toll-like receptor; BBB—blood–brain barrier.

**Table 1 ijms-23-02013-t001:** The effects of stress on gut and brain pathology and behaviour in male animal models of stress and depression.

Stress Model	Phylum				Sample Site
	Firmicutes	Bacteroidetes	Actinobacteria	Proteobacteria	
CUMS	Increased [21,42,51,52,54,64,68,106]	Increased [21,95,104,107,108,109]	Increased [34,42,52,54,108]	Increased [21,108]	Cecum [51,106]
	Decreased [63,95,104,107,108,109]	Decreased [42,51,52,54,64,68]	Decreased [104]	Decreased [42]	Faecal pellets [21,42,52,54,63,64,68,95,103,104,107,108,109,110]
	No change [110]	No change [103,106,110]	-	No change [54]	
CRS	-	Increased [69,98,99]	-	Increased [69,96,111]	Cecum [69,99,111]
	Decreased [98,99]	-	Decreased [96,98]	-	Faecal pellets [28,96,98,112,113]
	No change [28,112,113]	No change [28,34,112,113]	No change [28,113]	No change [28,113]	
MS	-	-	-	-	-
CORT	Increased [62,114]	Decreased [62,114]		Increased [114]	Cecum [114]
CSDS	Decreased [76]	Increased [76]	Increased [76]	Increased [76,115]	Cecum [100,115]
	-	-	Decreased [115]	Decreased [100,116]	Faecal pellets [76,117]
	No change [100,116,117]	No change [100,116]	-	-	Colonic content [116]
LH	-	-	-	-	-

CUMS—chronic unpredictable mild stress; CRS—chronic restraint stress; MS—maternal separation; CORT—chronic corticosteroid administration; CSDS—chronic social defeat stress; LH—learned helplessness; -, not investigated.

**Table 2 ijms-23-02013-t002:** Comparisons of gut microbiota alterations at the genus level and the site of sample collection across male animal models of stress and depression.

Stress Model	Genus						Sample Site
	*Lactobacillus*	*Bacteroides*	*Clostridium*	*Bifidobacterium*	*Allobaculum*	*Turicibacter*	
CUMS	Increased [64,68]	Increased [21,68,104]	Increased [68]	Increased [42,108]	Increased [42,54]	-	Cecum [51]
	Decreased [21,52,108,109]	Decreased [108,110]	Decreased [63]	Decreased [104]	-	Decreased [54]	Faecal pellets [21,42,52,54,63,64,68,103,104,108,109,110]
	-	-	No change [21]	-	-	No change [21]	
CRS	Increased [125]	Increased [99,120]	-	-	-	-	Cecum [69,99]
	Decreased [69,92,99]	Decreased [69]	Decreased [112]	Decreased [112]	Decreased [112,113]	Decreased [112]	Faecal pellets [112,113,120,125]
							Mid-colonic section [92]
MS	Increased [23]	Increased [101,121]	-	Increased [23]	-	-	Faecal pellets [23,83,97,101,121,126]
	Decreased [97]	Decreased [97]	Decreased [101,121]	-	-	Decreased [101]	
	-	No change [83]		-	-	-	
CORT	-	-	-	Decreased [67]	Decreased [114]	-	Cecum [114]
							Faecal pellets [67]
CSDS	-	-	Increased [117]	-	-	-	Cecum [100,115,127]
	Decreased [116]	-	-	Decreased [115]	Decreased [115,116]	Decreased [100]	Faecal pellets [117]
	-	No change [100,127]	No change [100]	No change [116]	-	-	Colonic content [116]
LH	Increased [94]	-	Increased [94]	-	-	-	Faecal pellets [94,128]
	No change [128]		-				

CUMS—chronic unpredictable mild stress; CRS—chronic restraint stress; MS—maternal separation; CORT—chronic corticosteroid administration; CSDS—chronic social defeat stress; LH—learned helplessness; -, not investigated.

**Table 3 ijms-23-02013-t003:** The effects of diet on gut pathology, behaviour and potential mechanisms for the brain in animal models of stress and depression.

Model, Duration and Species	Diet/Treatment	Gut Pathology	Behaviour	Possible Gut–brain Pathways	Authors
Chronic unpredictable mild stress (CUMS)—4 weeks Male Wistar rats	Standard diet + Orally gavaged fructo-oligosaccharides (FOS)/ galacto-oligosaccharides (GOS) and probiotics (*Bifidobacterium longum* and *Lactobacillus rhamnosus*)	Colonic serotonin (5-HT) and tryptophan hydroxylase 1 (TPH1); Attenuated by FOS/GOS and probiotics	Depressive-like behaviour; Attenuated by FOS/GOS and probiotics Anhedonia; Attenuated by FOS/GOS and probiotics	Enteroendocrine alterations and perturbations in tryptophan metabolism	[51]
CUMS—4 weeks C57BL6 mice of unspecified sex	Standard diet + orally gavaged *Bifidobacterium longum* subspecies infantis strain CCFM687 in 10% skimmed milk solution at 10^9^ CFU/mL daily for 6 weeks	Total short chain fatty acids (SCFA); Attenuated by CCFM687	Depressive-like behaviour; Attenuated by CCFM687 Anxiety-like behaviour; Attenuated by CCFM687	Alterations to SCFA regulation of intestinal permeability affected systemic inflammation and HPA axis function, leading to changes in neuroplasticity in the frontal cortex	[54]
CUMS—4 weeks Male Sprague Dawley rats	Standard diet + orally gavaged Fluoxetine (1.82 mg/kg), green tea (64.8 mg/kg) or jasmine tea (21.6 mg/kg, 64.8 mg/kg and 194.4 mg/kg)	Colonic structural integrity (inflammatory infiltration, decreased goblet cell number and shallow crypts); Attenuated by jasmine tea	Depressive-like behaviour; Attenuated by Fluoxetine, green tea and jasmine tea	Alterations in peripheral (glucagon-like peptide 1) GLP-1 release in the gut with subsequent alterations in vagal-dependent central GLP-1 signalling in the brain	[68]
CUMS—4 weeks Male C57BL6 mice	Standard diet + orally gavaged saline, partially hydrolysed guar gum (PHGG) (600 mg/kg)), Fluoxetine (0.5 mg/kg) and PHGG (600 mg/kg) or Fluoxetine (1.0 mg/kg)	Faecal SCFAs (lactic acid, acetic acid and valeric acid); Attenuated by PHGG, PHGG + Fluoxetine and Fluoxetine	Depressive-like behaviour; Attenuated by PHGG, PHGG + Fluoxetine and Fluoxetine	Alterations to SCFA regulation of intestinal permeability leading to changes in serotonergic and dopaminergic neurotransmission in the striatum and hippocampus	[63]
CUMS—5 weeks Male C57BL6 mice	Standard diet + orally gavaged CCFM105 *Bifidobacterium breve* (0.1 mL/10 g body weight)	Colonic serotonin (5-HT) and tryptophan hydroxylase 1 (TPH1); Attenuated by *B.breve* Colonic SCFAs (propionate, butyrate, isobutyric acid and isovaleric acid); All attenuated by *B. breve*	Anhedonia; Attenuated by *B. breve* Depressive-like behaviour; Attenuated by *B. breve* Anxiety-like behaviour; Attenuated by *B.breve*	Alterations to SCFA regulation of intestinal permeability and colonic enzymes involved in serotonin synthesis affected 5-hydroxytryptophan (5-HTP) levels. 5-HTP is capable of crossing the BBB, thus these alterations influenced neuroplasticity in the hippocampus. Alterations in HPA axis function may have also contributed the changes in brain and behaviour	[42]
CUMS—7 weeks Male C57BL6 mice	Standard diet or standard diet + glycated milk casein (Gc) or glycated milk casein fermented with *Lactobacillus rhamnosus* (FGc)	Colonic tryptophan hydroxylase 1 (TPH1) and free fatty acid receptor 2 (GPR43); Attenuated by FGc Colonic inflammation (iNOS and COX-2); iNOS attenuated by FGc and COX-2 attenuated by Gc and FGc Colonic barrier integrity (Zo-1, Cldn-5 and occludin); Attenuated by Gc and FGc bar occludin	Anxiety-like behaviour; Attenuated by FGc	Alterations in colonic inflammation may modulate colitis pathology, affecting intestinal permeability and this may affect the transport of gut-derived molecules such as inflammatory mediators in circulation which can reach the brain. These alterations, which may also involve the HPA axis, may affect neuroplasticity in the brain	[21]
Chronic restraint stress (CRS)—1 week Male C57BL6 mice	Standard diet + orally gavaged saline or *Lactobacillus johnsonii*	Jejunal and ileal barrier integrity (Zo-1, Cldn-1 and occludin); Attenuated by *L. johnsonii* Ileal inflammation (TNF- α and INF-γ); Attenuated by *L. johnsonii*	Memory; Attenuated by *L. johnsonii*	Alterations in small bowel inflammation may modulate intestinal barrier integrity subsequently impacting hippocampal dopaminergic, serotonergic and GABAergic neurotransmission	[66]
CRS—2 weeks Male Swiss mice	Standard diet + orally gavaged *Bifidobacterium longum* BG0014, *Bifidobacterium longum ssp. infantis* Bl11471, *Bifidobacterium animalis* BL0005, *Bifidobacterium animalis ssp. lactis* 420, *Lactobacillus paracasei* Lpc-37, *Lactobacillus salivarius* Ls-33, *Lactobacillus plantarum* LP12418, *Lactobacillus plantarum* LP12151, *Lactobacillus plantarum* LP12407, *Lactobacillus acidophilus* LA11873, *Lactobacillus rhamnosus* LX11881 or *Lactobacillus helveticus* LH0138 (1 × 10^9^ CFU/day for 5 weeks)	No significant changes to ${\mathrm{GABA}}_{B}$ receptor mRNA expression; No significant effects of probiotics	Anxiety-like behaviour; Attenuated by Lpc-37, LP12407 and LP12418 Depressive-like behaviour; Attenuated by Lpc-37, LP12407 and LP12418 Memory; Attenuated Lpc-37, LP12407 and LP12418	Alterations in behaviour modulated by changes in gut microbiota with impacts on GABAergic neurotransmission in the prefrontal cortex requiring further study	[74]
Chronic social defeat stress (CSDS)—10 days Male C57BL6 mice and male CD1 mice	Standard diet + *Clostridium butyricum* MIYAIRI 588 (>5 × 10^6^/CFU) in drinking water for 4 weeks	Colonic inflammation (TNF-α, IL-1β and IL-6); MIYAIRI 588 attenuated all Colonic barrier integrity (Zo-1); Attenuated by MIYAIRI 588 Colonic free fatty acid receptors 2 and3; Attenuated by MIYAIRI 588	Depressive-like behaviour; Attenuated by MIYAIRI 588 Social interaction; Attenuated by MIYAIRI 588	Alterations in neuroinflammation and colonic inflammation modulated by alterations in the gut microbiota and intestinal barrier integrity	[77]

CUMS—chronic unpredictable mild stress; CRS—chronic restraint stress; MS—maternal separation; CORT—chronic corticosteroid administration; CSDS—chronic social defeat stress; LH—learned helplessness; HPA—hypothalamic-pituitary-adrenal axis; BBB—blood brain barrier.

## Data Availability

Not applicable.

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
