# Peer review of "The Effects of Stress and Diet on the “Brain–Gut” and “Gut–Brain” Pathways in Animal Models of Stress and Depression"

_ijms, 2022, doi:10.3390/ijms23042013_

Round 1
Reviewer 1 Report
A general and appropriate review of the gut-brain axis. Comprehensive and illuminates issues---reasonably novel
Author Response
The authors would like to thank Reviewer 1 for their positive review of the manuscript. Since the gut-brain axis is a trending point of discussion in the neurosciences, the manuscript has been updated with recent key references to enhance novelty and comprehension.
Reviewer 2 Report
This is very interesting topic. The connection between gut and brain that can contribute to neuropsychiatric disorders such as depression is an emerging area of the microbiome study, and this can impact the future treatment methods for other neuropsychiatric disorders. The manuscript is very well written and well organized. I could only find a very few things that may need edits.
- The figure 1 representation is not clear. Authors can use different types of arrows to show positive or negative effects and additional labeling of the arrows to explain the activities caused by stress or gut microbiota alteration.
- Regarding lines 196 and 198, Could you put the numbering on the subtopics? It will be helpful for readers.
- For lines 509-513, It needs to be more cautious about saying that gut microbiota composition is associated with depressive-like behavior because the fecal transplantation not only transfers bacteria but also transfers body fluid, metabolites hormones, etc.
Author Response
The authors would like to thank Reviewer 2 for their review of the manuscript. Please find the responses to each suggestion below in blue:
1. The figure 1 representation is not clear. Authors can use different types of arrows to show positive or negative effects and additional labeling of the arrows to explain the activities caused by stress or gut microbiota alteration.
The authors agree that Figure 1 requires more clarity. This figure has now been updated to better facilitate readers’ understanding of the overview of the major brain-to-gut and gut-to-brain pathways.
2. Regarding lines 196 and 198, Could you put the numbering on the subtopics? It will be helpful for readers.
Numbering on the subtopics referenced has been added to better facilitate readers as suggested.
3. For lines 509-513, It needs to be more cautious about saying that gut microbiota composition is associated with depressive-like behavior because the fecal transplantation not only transfers bacteria but also transfers body fluid, metabolites hormones, etc.
This is an excellent suggestion. We have updated this statement with more recent references and the wording has been adjusted to make it clear to readers that the effects seen with fecal transplantation may not be solely due to bacterial diversity or composition.
Reviewer 3 Report
In this review, Mauritz Herselman et al. discussed the effects of stress and diet on the bidirectional communication between brain and gut. HPA Axis, Neural Pathways and Serotonin and Tryptophan Pathways are important for brain to gut communication. Intestinal Barrier Integrity Pathways, and Gut Microbiota Pathways contribute to “gut to brain“ communication. They also discussed the effects of diet on brain-gut pathway in several models for depression. This review is novel, clear, and comprehensive. I think it will promote the investigation for the research on the gut-brain communication in depression and related diseases.
Author Response
The authors would like to thank Reviewer 3 for their positive review of the manuscript. Since the gut-brain axis is a trending point of discussion in the neurosciences, the manuscript has been updated with recent key references to enhance novelty and comprehension.
Reviewer 4 Report
Dear Editor, 20220123
The present manuscript Mauritz Herselman and coworkers aim to provide a review of the “latest research on the effects of stress on the bidirectional connections between the brain and the gut across the most widely used animal models of stress and depression” by focusing on the diversity and composition of the gut microbiota.
Many publications, especially during the last years, have added to our understanding of the inter-relationship between the gut microbiome/metabolites produced by the microbita and the brain functions, including stress and depression.
The manuscript is well written, but in my opinion, has neglected some crucial recently published articles. In other words, the topic is really interesting but the selection of literature reviewed by the authors is incomplete to provide a comprehensive picture. To my opinion, the authors should consider -at least- including information given in the suggested sources below into their article to be seriously considered for publication in Int. J. Mol. Sci.
-Ann Nutr Metab, 2021;77 Suppl 2:11-20., doi: 10.1159/000518274. Gut Microbiota and Pathophysiology of Depressive Disorder
Hiroshi Kunugi
-Nutrients, . 2021 Feb 25;13(3):732., doi: 10.3390/nu13030732. The Role of Gut Bacterial Metabolites in Brain Development, Aging and Disease
Shirley Mei-Sin Tran et al.,
-Food Funct, . 2021 May 21;12(10):4284-4314., doi: 10.1039/d0fo02855j. Gut microbiota in mental health and depression: role of pre/pro/synbiotics in their modulation
Hasnain N Methiwala et al.,
-Eur J Neurosci, . 2021 Apr 20., doi: 10.1111/ejn.15239. Inflammation-driven brain and gut barrier dysfunction in stress and mood disorders
Ellen Doney et al.,
-Nutrients, 2021 Dec 23;14(1):37., doi: 10.3390/nu14010037. The Role of the Gut Microbiota in the Development and Progression of Major Depressive and Bipolar Disorder
Tom Knuesel et al.,
-Metabolites, . 2022 Jan 8;12(1):50., doi: 10.3390/metabo12010050. Gut Microbiota Metabolites in Major Depressive Disorder-Deep Insights into Their Pathophysiological Role and Potential Translational Applications Miguel A Ortega , et al
Author Response
The authors would like to thank Reviewer 4 for their thorough review of the manuscript. It is agreed that the manuscript lacked some recent important findings in the inter-relationship between the gut and the brain in stress and depression. The manuscript has been updated to provide a more up-to-date and comprehensive picture.
Reviewer 5 Report
This is a comprehensive review of the ways in which stress and diet impact the connection between the gut and the brain in animal models of mood disorders. I have the following suggestions that I believe are needed and will improve this review.
- Lines 72-78 – As I think there will be readers of this paper for whom modelling neuropscyhiatric disorders is somewhat novel, it would be nice to begin this paragraph with an acknowledgement of how challenging it is to model a human disorder like depression in animals and the associated limitations. The authors could point out that the best approach is to model individual behaviours that are disrupted in depressed patients – e.g., anhedonia; and then give some explanation as to how such a behaviour is measured in rodents - e.g., measuring the preference for a sweet substance.
-
In paragrpaphs from lines 72-113, there is a lot of dependence on ref#8 to the point that primary research articles are not being cited when appropriate. These paragraphs from provide a wealth of information about the animal models but very few primary references are used and as a result, there is a lack of detail provided.
-
With further respect to the paragraphs from lines 72-113, the authors are encouraged to be more specific with the information they provide. For example, lines 80-87 -this seems an oversimplification of the role of HPA axis dysregulation - HPA axis dysregulation in depression can also include hypoactivation, not just hyperactivation. As the authors add more primary references as per Point#2, these paragraphs can be fleshed out to include more and accurate details.
- Despite the fact that mood disorders are diagnosed more in women relative to men, there remains a continued neglect by authors to include sex as a factor in experiments in which animal models of these disorders are used. This review needs to be edited throughout to refer to sex of subjects when citing results of past papers, and especially paragraphs of lines 72-113 to include a discussion of sex differences for each topic. It may be argued that sex has not been well reported in the gut-brain literature, but sex differences in stress and mood disorders have been known for years.
-
Table 3 – The sex of animals used in the various studies needs to be included whether or not the results were different in the sexes. If they were, the results for each sex should be provided.
Author Response
This is a comprehensive review of the ways in which stress and diet impact the connection between the gut and the brain in animal models of mood disorders. I have the following suggestions that I believe are needed and will improve this review.
The authors would like to thank Reviewer 5 for their thorough review of the manuscript. Please find the responses to each of the points below in blue.
1. Lines 72-78 – As I think there will be readers of this paper for whom modelling neuropscyhiatric disorders is somewhat novel, it would be nice to begin this paragraph with an acknowledgement of how challenging it is to model a human disorder like depression in animals and the associated limitations. The authors could point out that the best approach is to model individual behaviours that are disrupted in depressed patients – e.g., anhedonia; and then give some explanation as to how such a behaviour is measured in rodents - e.g., measuring the preference for a sweet substance.
This paragraph has been updated to better accommodate readers for whom modelling psychiatric disorders is relatively novel. A summary of commonly used behavioural tests and their purposes has been included too.
2. In paragrpaphs from lines 72-113, there is a lot of dependence on ref#8 to the point that primary research articles are not being cited when appropriate. These paragraphs from provide a wealth of information about the animal models but very few primary references are used and as a result, there is a lack of detail provided.
Please see point 3.
2. With further respect to the paragraphs from lines 72-113, the authors are encouraged to be more specific with the information they provide. For example, lines 80-87 -this seems an oversimplification of the role of HPA axis dysregulation - HPA axis dysregulation in depression can also include hypoactivation, not just hyperactivation. As the authors add more primary references as per Point#2, these paragraphs can be fleshed out to include more and accurate details.
More primary references have been included to facilitate a more detailed accurate discussion of animal models of stress and depression and the biological mechanisms involved.
4. Despite the fact that mood disorders are diagnosed more in women relative to men, there remains a continued neglect by authors to include sex as a factor in experiments in which animal models of these disorders are used. This review needs to be edited throughout to refer to sex of subjects when citing results of past papers, and especially paragraphs of lines 72-113 to include a discussion of sex differences for each topic. It may be argued that sex has not been well reported in the gut-brain literature, but sex differences in stress and mood disorders have been known for years.
This is an excellent suggestion. While animal studies in females are currently lacking in the gut-brain literature, the manuscript has been edited to make known sex differences in stress and depression clearer to readers. Recent publications regarding female animals in models of stress and depression have also been included throughout the paper.
5. Table 3 – The sex of animals used in the various studies needs to be included whether or not the results were different in the sexes. If they were, the results for each sex should be provided.
Table 3 has been updated with a few more recent publications and the sex of the animals used in each study has been made clear in the table as well as throughout the rest of the manuscript.
Round 2
Reviewer 4 Report
Thanks for your revisions. My concerns are mostly taken care of.
Reviewer 5 Report
The authors did an excellent job taking my concerns into consideration and revising the manuscript accordingly. I feel strongly that such contributions should include reference to both sexes and the authors embraced that suggestion and the result is a more comprehensive review. Thank-you.